# Production of Probiotic Passion Fruit (*Passiflora edulis* Sims f. *flavicarpa* Deg.) Drink Using *Lactobacillus reuteri* and Microencapsulation via Spray Drying

**DOI:** 10.3390/foods9030335

**Published:** 2020-03-12

**Authors:** Shênia Santos Monteiro, Yolanda Albertina Silva Beserra, Hugo Miguel Lisboa Oliveira, Matheus Augusto de Bittencourt Pasquali

**Affiliations:** Unidade Academica de Engenharia de Alimentos, Universidade Federal Campina Grande, Av. Aprígio Veloso 882, Campina Grande 58429-200, Brazil; shenia-monteiro@hotmail.com (S.S.M.); yolanda.beserra@hotmail.com (Y.A.S.B.); hugom.lisboa80@gmail.com (H.M.L.O.)

**Keywords:** encapsulation, probiotic, *Passiflora edulis* Sims f. *flavicarpa* Deg., *Lactobacillus reuteri*

## Abstract

Probiotic foods offer many benefits to human health, causing increased interest in the development of new food products that exploit such benefits. However, traditional dairy foods are being replaced by other non-dairy foods to provide additional sources of benefits provided by bioactive molecules. Therefore, the objective of the present work was to study the production process of a probiotic fruit drink and then microencapsulate the probiotic pulp to stabilize the drink further. Passion fruit pulp (*Passiflora edulis* Sims f. *flavicarpa* Deg.) was fermented with *Lactobacillus reuteri* under different temperature conditions in combination with different pHs to find the best fermentation conditions. Different from dairy sources, the optimal conditions for the growth of *Lactobacillus reuteri* in the passion fruit pulp were found to be 30 °C at pH 3.18, where phenolic compounds could also be used as a secondary metabolic pathway. Spray-drying was performed using different conditions for microencapsulation. Process yields and *Lactobacillus reuteri* survival showed the dependency of droplet sizes, whereas phenolic compound retention was increased when higher amounts of gelatin were used. Therefore, the development of a new food product comprising a powdered fruit pulp rich in probiotic and phenolic compounds was possible.

## 1. Introduction

In recent years, increasing attention has been given to functional foods, mainly due to the interest in consuming foods that have beneficial health properties, such as probiotic foods [1]. Probiotic cultures provide several health benefits that are still under-researched in order to be fully exploited [2]. However, maintenance of healthy intestinal microbiota; protection against gastrointestinal pathogens; increasing the immune system; and reduction of serum cholesterol level, blood pressure, and anticancer activity are some of the benefits already reported [3].

Nowadays, considerable attention is being given to extending probiotics into non-dairy foods. The motivation is related to groups of people who generally do not consume them for any reason, such as lactose intolerance, milk allergies, or simple dislike [4]. Thus, the search for different products opens an opportunity to combine the benefits of probiotic cultures with bioactive molecules. However, the addition of probiotic on non-dairy products can be challenging, especially in fruit juices because of low pH, organic acids, and the absence of molecules typically present on milk. Although substantial challenges remain, non-dairy probiotic foods have already been reported, such as fruit juices [5]; legume juice [6]; soy, almond, and coconut milk [7]; and cereal-based milk [8].

Tropical fruits are an excellent source of vitamins, minerals, and antioxidants, and are highly appreciated worldwide. Among the tropical fruits, yellow passion fruit (*Passiflora edulis* Sims f. *flavicarpa* Deg.) is a fruit of wide acceptability among populations around the world. Originating in tropical America, it is widely cultivated in Brazil. However, one of the problems identified by the production chain and commercialization is the reduced shelf-life of the fruit, impacting its external appearance, in which the loss of mass and consequently the consistency gives a wrinkled appearance to the fruit [9,10].

Dehydrated and powdered food products are widely used in the food industry because of the increased shelf-life and easiness for use as additives on other food products. Spray drying is a commonly used technique on an industrial scale for food dehydration [11]. Consequently, when applied to fruits, preservation of bioactive molecules becomes of fundamental importance. Spray-drying or encapsulation of probiotic products has been previously reported [12]. In a recent study, it was reported that the wall material concentration influenced the encapsulation of a probiotic culture [13]. In terms of probiotics, researchers have optimized the spray-drying process optimized for powdered Moringa (*Moringa oleifera* Lam) leaf supporting different probiotic cultures [14]. The storage of *Bifidobacterium* encapsulated on passion fruit was studied after spray-drying [5].

Most of the literature on spray-drying on fruit products focuses on the optimization of process conditions. The reason behind this is due to the presence of low molecular weight sugars and organic acids, which makes the spray-drying process problematic because of the low glass transition of the powder. Typically the strategy used is to include encapsulating agents such as maltodextrin and gum arabic, among others. However, the combination of high cell viability with high bioactive compound retention can also be challenging for a spray drying process. Therefore, the objective of our work was to produce a powdered probiotic from fermentation of *Lactobacillus reuteri* on yellow passion fruit (*Passiflora edulis* Sims f. *flavicarpa* Deg.) pulp using spray-drying.

## 2. Materials and Methods

### 2.1. Materials

The yellow passion fruit (*Passiflora edulis* Sims f. *flavicarpa* Deg.) and gelatin were purchased at the local market of Campina Grande (Paraíba Brazil). The culture of *Lactobacillus reuteri* DSM 17938 was purchased from the company Official Pharmacy (São Roque, Brazil). Maltodextrin (DE 10) was purchased from INGREDION BRASIL-Ingredientes Industriais (São Paulo, Brazil).

### 2.2. Passion Fruit Pulp

To obtain the passion fruit pulp, the fruits were pre-selected, taking into account characteristics such as color, physical damage, stage of maturation, peel quality, and size. Afterward, the fruits were washed in running water to remove dirt from the market, and were sanitized in 200 ppm sodium hypochlorite solution for 30 min. The fruits were cut manually with the aid of a stainless steel knife. Then, they were stripped and the seeds were dismembered using a mesh #10 sieve. The pulp was packed in a plastic container and stored at −20 °C until analysis.

### 2.3. Growth Kinetics

Probiotic pulp was obtained using 433 mL of the passion fruit (*Passiflora edulis* Sims f. *flavicarpa* Deg.) pulp for each experiment (pH 3.18, pH 5.5, pH 6.5). Then, the pH of the samples was adjusted to 5.5 and 6.5 with NaOH solution (3M). The lyophilized probiotic strain was added to the pulp to obtain a concentration greater than 106 CFU/mL and stirred for 10 minutes to obtain a homogeneous medium. The biological characteristics of *Lactobacillus reuteri* in passion fruit pulp were determined by monitoring the fermentation for 48 h at temperatures of 30, 20, and 10 °C. At each 2 hour interval, the cell concentration was determined by direct counting in Neubauer’s chamber, pH, total titratable acidity and total soluble solids [15], reductive carbohydrates [16], and phenolic compounds [17,18] (represented as equivante gallic acid—EGA). From the growth data of the microorganism in the passion fruit pulp, the maximum growth rate, the generation time, and the yield cell/substrate were calculated according to Equations (1), (2) and (3), respectively.
(1)$$\ln\frac{X}{X_{i}}{=\mu }_{m}\times \left({t-t}_{i}\right)$$
(2)$$\mu_{m}=\frac{\ln2}{t_{g}}$$
(3)$$Y_{cell/substrate}=\frac{\Delta X}{\Delta S}$$
where X is the cell concentration at the end of the exponential phase (cell/mL), $X_{i}$ is the concentration of cells at the beginning of the exponential phase (cell/mL), $\mu_{m}$ is the maximum growth rate (h^−1^), t is the time corresponding to the end of the exponential phase (h), $t_{i}$ is the time corresponding to the beginning of the exponential phase (h), $t_{g}$ = generation time (h), Ycell/substrate is the cell production yield to substrate, ΔX is the difference between X and Xi (cell/mL), and ΔS is the substrate consumption at the end of the exponential phase (g/mL).

### 2.4. Spray Drying

The spray drying of the probiotic passion fruit pulp was performed using a Labmaq SD10 spray dryer, equipped with a two-hole nozzle (2 mm hole), with atmospheric air as drying gas in open cycle mode (without air recirculation). Two liters of passion fruit pulp was fermented with *Lactobacillus reuteri* at a pH 3.18 at 30 °C for 12 h. Then, five samples of 400 mL were separated, and 60 g of maltodextrin DE10 was added. The experiments were conducted according to a 2² factorial design with four distinct points and three repetitions at the central point. Atomization air flow rates at two levels (2–4 kg/h) and gelatin concentration at two levels (0–2%) were used as factors. The drying process was performed with a drying air inlet temperature of 112 °C, and a product outlet temperature at 74 ± 2 °C. The drying air flow rate was 430 kg/h, and the product flow rate was 3 kg/h. The input temperatures of the drying air, product outlet temperature, drying air flow rate, and product flow rate were kept constant for all experiments performed.

#### 2.4.1. Process Yield

Process yield was calculated by the ratio between the mass of powder collected, expressed in grams, with the total solids fed to the spray drying. The latter was calculated by the probiotic fruit pulp solids fraction multiplying by the total weight of each sample fed to the spray drying (Equation (4)).
(4)$$Yield\left(\%\right)=\frac{Mass\;of\;powder\;collected\;in\;cyclone}{Total\;solidfraction*Sample\;Total\;Weight}\times 100$$

#### 2.4.2. Cell Number and Encapsulation Efficiency

Cell number after spray-drying and encapsulation efficiency were calculated by Equations (5) and (6).
(5)$$Log\;Cell\;Number=Log\;N_{0}-Log\;N$$
(6)$$E.E.\left(\%\right)=\frac{N}{N0}$$
where N_0_ is the number of cells in the feed solution (cell mL^−1^), and N is the number of cells in the powder, both measured by direct counting in Neubauer’s chamber.

#### 2.4.3. Powder Moisture Content

The powder moisture content was determined using gravimetric methods according to the Association of Official Analytical Chemists (AOAC) method 925.10 [19].

### 2.5. Statistical Analysis

Powder characterization results were assessed by one-way ANOVA followed by Tukey comparisons test at 5% of probability. The experimental results were fitted to a first-order polynomial equation (Equation (7)) by non-linear regression, and the models were verified using ANOVA for the same level of probability. Statistical calculations were performed using GraphPad Prism version 8.3.0 for macOS, GraphPad Software (San Diego, CA, USA).
(7)y=b0+b1∗x1∗b2∗x2+b3∗x1∗x2
where y is the response, x_1_ and x_2_ are the independent variables, and b_n_ are the regressed parameters.

## 3. Results and Discussions

### 3.1. Characterization of Passion Fruit Pulp in Natura

The composition of passion fruit pulp was determined to assess its potentiality to produce a probiotic product. The average values and standard deviation of the characterized parameters of the passion fruit pulp in natura are presented in Table 1. pH is a parameter that directly influences the fermentation process because microorganisms’ development and functions depend on this parameter. Fruit pulp extracted in our study presented a pH of 3.18, a similar value found for pasteurized and frozen passion fruit pulp [20]. The use of lactic acid bacteria for the development of new probiotic products have been reported [21]. In these terms, the *Lactobacillus reuteri* is regarded as a prominent microorganism targeted for new probiotic applications [21]. *Lactobacillus reuteri* have an optimal growth at pH 6.0 when cultivated in a commercial medium. Moreover, when submitted to other conditions of pH, *Lactobacillus reuteri* also present growth, although the time to reach the stationary phase is higher [22]. The pH measured in the passion fruit pulp extracted in our study was pH 3.18, which to *Lactobacillus reuteri* apparently could not be optimal for the microorganism growth when compared with optimal pH studied in others’ works [22]. However, it is important to highlight the possibility of the use of passion fruit pulp as a medium for *Lactobacillus reuteri* growth, principally because it considered an innovative product.

In terms of physicochemical compositions, we quantified the concentration of reducing sugar, protein, phenolic compounds, and titratable acidity. It is known that the presence of free sugars in the pulp favors bacterial growth, which utilizes sugar as the main energy source. The reducing sugar content found on passion fruit pulp was 4.86%. This value is similar to the concentration of lactose content found on whole bovine milk, which is frequently used to ferment and growth of *Lactobacillus reuteri* culture [23]. Is important reinforce that reducing sugar present on passion fruit pulp is different from reducing sugar found in milk. In this way, is known that *Lactobacillus reuteri* can metabolize sugars that are different from lactose as an alternative energy source. Additionally, Filanino and co-workers, while studying the metabolism of different *Lactobacillus sp*. during the lactic acid fermentation of fruit and vegetables, found that phenolic compounds can also offer an alternative metabolic route that provides an energetic advantage for lactic acid bacterial growth [24].

The phenolic content measured in the pulp of passion fruit in natura extracted in our study was similar to that found by Oliveira et al. [25], which presented levels of phenolic compounds of 32.40 mg EGA/100 g and 27.10 to 36.47 mg EGA/100 g. In terms of this, it was recently reported that a high content of phenolic compounds influences the growth and the survival of lactic acid bacteria such as *Lactobacillus plantarum* in different fruit juices [26]. Additionally, authors also have discussed the importance that other components, such as proteins and fibers, play in terms of the development and growth of lactic acid bacteria cultures. The in natura pulp extracted of the passion fruit presented an average protein content of 0.62 mg/mL of pulp. Similar values were found by Araújo et al. [27], finding a protein content for passion fruit of 0.8 mg/mL of pulp. The variation in the protein profile content of the passion fruit pulp may occur due to different factors in terms of the genotype and stage of fruit maturation. In terms of this, it is important to standardize the maturation stage of the passion fruit for exztraction purposes, principally for innovative use on probiotic products. The total titratable acidity measured in the passion fruit pulp extracted was 4.55%. The titratable acidity plays an important role in the probiotic products due to the influence in both the physicochemical characteristics and microorganism development [28]. Additionally, the presence of natural organic acids present in fruit juices and fruit pulp can be related to unpleased taste, consequently interfering in the consumer acceptability. Regarding *Passiflora spp*. Rotta and co-workers found the presence of 4-hydroxybenzoic acid, chlorogenic acid, vanillic acid, caffeic acid, *p*-coumaric acid, ferulic acid, and *trans*-cinnamic acid on different passion fruit pulps, including *Passiflora edulis* Sims f. *flavicarpa* Deg. fruit [29]. These compounds are related to antioxidant and organoleptic properties, which are also related to health benefits and food product quality.

### 3.2. Lactobacillus Reuteri Growth during Passion Fruit Pulp Fermentation

Passion fruit pulp extracted from *Passiflora edulis* Sims f. *flavicarpa* Deg. was fermented using *Lactobacillus reuteri* under three different pHs (3.18, 5.5 and 6.8) at three different temperatures (10, 20 and 30 °C). Figure 1 presents the effects of both variables on cell concentration, reducing sugars, and phenolic content during the fermentation period. The lag phase of all sets of fermentation presented an increased lag period when the pH was 5.5 and 6.5. Interestingly, these pH were closer to the optimum pH for the growth of *Lactobacillus reuteri*. However, it is important to emphasize that optimum pH reported for *Lactobacillus reuteri* was using commercial medium to fermentation [22]. In these terms, the physicochemical characteristic of passion fruit pulp used here to fermentation were different of the characteristics of the milk, which is usually used to growth *Lactobacillus reuteri*. Therefore, we suggest that the increase in the lag phase observed in our study may have been caused by the addition of hydroxide sodium liquor to adjust the pH of the batch for fermentation before the inoculation. Wang and co-workers, studying the lactic acid production with *Lactobacillus casei,* found increase in the lag phase after using hydroxide sodium liquor to ajust the pH of fermentation medium [30]. In terms of this, we belive that the addition of hydroxide sodium liquor may have altered the structure of compounds present on the passion fruit pulp. As a consequence, this resulted in an increased adaptation time of *Lactobacillus reuteri* to this medium.

The influence of temperature and pH on maximum cell concentration, generation time, and maximum growth can be observed in Figure 2. Maximum cell concentration was found at the temperature of 30 °C (Figure 1) in combination with pH 3.18 when compared with pH 6.5 and pH 5.5 at the same temperature. The lowest cell concentration was found at the temperature of 10 °C and at pH 3.18. In our study, we designed a set of batch conditions to investigate and optimize the production of an innovative probiotic product, using both passion fruit pulp and *Lactobacillus reuteri.* In terms of lactic acid bacteria specifically, *Lactobacillus reuteri* is known as a culture that presents optimal growth between 25 and 40 °C, and that the growth development is dependent upon fermentation medium [31]. Here, for the first time, the passion fruit pulp (*Passiflora edulis* Sims f. *flavicarpa* Deg) was used as a medium for fermentation of *Lactobacillus reuteri.* In our design of conditions, we observed that the optimal combined parameters for pH and temperature were 3.18 and 30 °C, respectively. This evaluation of the impact that different variables make on cell development is fundamental to optimize the process of fermentation and make it easily applicable to the food industry. Moreover, it is important to mention that the conditions here found as optimal for *Lactobacillus reuteri* growth help in keeping the beneficial properties of the passion fruit pulp extracted.

In the set of batch fermentation here studied, we also calculated the highest maximum growth (μmax) rates. Interestingly, the highest maximum growth rates found for samples fermented were at 20 °C and pH 5.5, followed by 10 °C and pH 6.5 0.097 h^−1^ and 0.055 h^−1^, respectively. At the specific growth rate, the exponential phase was also characterized by the generation time (t*_g_*), which is the time interval needed to double the cell concentration in the medium. It was observed that the sample with pH 3.18 when fermented at 30 °C showed the shortest generation time, followed by samples with pH adjusted to 5.5 at 20 °C and pH 6.5 at 10 °C. Although *Lactobacillus reuteri* showed excellent growth in the samples with pH 3.18 at 30 °C and 20 °C, the generation time (t*_g_*) was greater than 2 h. Under optimal conditions, bacteria are the microorganisms with the highest growth rate and may have a generation time (t*_g_*) of less than 1 hour. Hernandez and co-workers found higher growth rates and lower generation times when *Lactobacillus reuteri* DSM 17938 was cultured under commercial medium [31]. In terms of this, it is important to recognize that our study had as its aim the development and optimization of variables in the passion fruit pulp extracted from *Passiflora edulis* Sims f. *flavicarpa* Deg. for the development of a probiotic product. The challenge in our study was to obtain a probiotic through the use of natural mediums using fruit pulp in an attempt to increase the health beneficial use of the probiotic produced.

### 3.3. Physicochemical and Phytochemical Characteristics of the Fermentative Process

Reducing carbohydrate content during the growth kinetics of *Lactobacillus reuteri* in passion fruit pulp is presented in Figure 1. A decrease in reducing carbohydrate content was expected because the main metabolic pathway of *Lactobacillus reuteri* uses monosaccharides such as fructose, glucose, and galactose for energy production, and consequently lactic acid. The highest substrate consumption was found at pH 6.5 for temperatures of 10 and 30 °C, followed by pH 5.5 at 30 °C. A similar result was also observed for the samples with pH 5.5 and 6.5 at the temperature of 30 °C. However, cell yield in terms of consumed reducing carbohydrate (Y_cell/reducing carbohydrate_) presented a maximum value of 11.08 Log cells/g of reducing carbohydrate and was found to be at 30 °C, pH 3.18. In this way, it is known that when conditions that do not represent an optimal physiological are used, stress is induced to cells, and this modulates the level of metabolism directly. Therefore, in stress conditions, bacteria can present complex mechanisms to react to different conditions of the environment, altering the metabolism, which induces a behavior of consuming a high content of substrates for the maintenance of cellular functions [32].

The concentration of phenolic compounds during the growth kinetics of *Lactobacillus reuteri* in passion fruit pulp at different pH levels (pH 3.18, pH 5.5, and pH 6.5) and temperatures of 30, 20, and 10 °C are illustrated in Figure 1. The lowest alterations in phenolic compounds was found for pH 3.18 at temperatures of 30 and 20 °C. A similar analysis was performed for the phenolic alteration, wherein a cell yield was calculated in terms of alterations in phenolic compounds (Y_cell/phenolic content_). A maximum value of 10.68 Log cells/g phenolic content was found for 30 °C, pH 3.18. Similarly, the phenolic compound’s alteration was also lower for higher cell production. However, in this case, it is important to report that the addition of hydroxide sodium liquor to adjust the pH of the batch for fermentation before the inoculation could also alter the molecular structures of the phenolic compounds and therefore caused alterations on this analytical measurement. According to Copello Rotili et al. [33], phenolic compounds are substances of secondary plant metabolism with different physiological functions. The phenolic composition is determined by genetic and environmental factors, or due to post-harvest stresses involving temperature, transpiration, oxygen, and pathogens, but can be modified by oxidative reactions. In terms of fermentation, phenolic compounds can be consumed through microorganisms during the metabolic process or can be produced by microorganisms by specific cellular metabolic pathways [34]. Here, in our study, the maintenance of phenolic compounds in the probiotic was one of the objectives, mostly to keep the main properties of these compounds in the final product.

According to Patel [35] and Tripathi and Gire [36], pH is one of the main significant factors affecting the viability of probiotics. Fruit juices naturally have a low pH and a high level of organic acids. Thus, the combined action of the acidic environment and the intrinsic antimicrobial activity of the accumulated organic acids affect the growth of probiotic bacteria. Figure 3 shows the curves for the pH during growth kinetics with pH 3.18, pH 5.5, and pH 6.5 at the respective temperatures of 30, 20, and 10 °C. The pH of samples remained stable throughout the fermentation process at all temperatures, except at 30 °C, where a small decrease in pH was observed throughout the fermentation time, possibly due to the remarkable increase in the number of cells and metabolic products formation. However, when analyzing the samples after 48 hours of fermentation, a pH decrease was observed for all samples. This decrease may have occurred due to the decline of the viable cell population. Figure 3 presents the titratable total acidity parameter during fermentation. As expected, the acidity of the samples remained stable during the fermentation process, even at the best growth conditions. Consequently, these results suggest that both cell concentration and the produced lactic acid did not alter the initial characteristics of the pulp.

### 3.4. Production of Probiotic Passion Fruit Powder using Spray Drying

Spray-drying experiments were conducted using the passion fruit pulp fermented at the best conditions, 30 °C and pH 3.18 for 12 h. These conditions were chosen because both cell and phenolic compounds presented higher concentrations at these conditions. To provide enough sample for spray-drying, the fermentation was performed using 2 L of passion fruit pulp, which corresponded to an increase of fivefold when compared to the initial trials. After 12 h, the fermented pulp presented a cell concentration of 11.81 Log cells mL^−1^ (Table 2). A simple experimental design (2^2^ + 3) was used to assess the influence of two parameters on the properties of a probiotic passion fruit powder. A response surface was generated for each dependent variable, as presented in Figure 4.

#### 3.4.1. Concentration of Viable Cells in the Passion Fruit Probiotic Pulp

A higher concentration of viable cells was obtained in sample 1, with 11.71 Log cells mL^−1^, and the lowest corresponded to sample 4. When compared to the population of *Lactobacillus reuteri* in passion fruit pulp, the viability varied between 0.09 Log to 0.4 Log. Similar results were found by Guerin et al. [37], where a reduction of 0.1 to 0.5 Log of the concentration of *Lactobacillus rhamnosus* GG encapsulated by spray-drying in milk matrix was observed. Encapsulation efficiency can be calculated by the ratio between the viable cells and the initially viable cells. The results revealed encapsulation efficiencies ranging from 99% to 96%. A first-order polynomial model was fitted to the experimental data (Equation (8)) to better understand the impact of process variables on the powder properties.
(8)$$\mathrm{Log}N=11.93-{0.137\times F}_{\mathrm{atom}}-0.076\times G+{0.013\times F}_{\mathrm{atom}}\times G,\;R^{2}=66.19\%$$
where Log N is the logartimic of cell concentration (Log cells mL^−1^), $F_{\mathrm{atom}}$ is the atomization air flow rate (kg/h), and G is the gelatin concentration (%).

The atomization air flow rate coefficient had a significant effect (*p* < 0.05) on the concentration of viable cells after the drying process. Both the atomization air flow rate and the gelatin concentration presented a negative effect on cell viability. Because the atomization air flow rate determines the droplet size, our result suggested that larger droplets provided better protection of cells, and thus cell viability was higher. The higher the atomization flow rates, the higher the kinetic energy available for the rupture of the liquid, resulting in smaller droplets [38]. Once the droplet size became smaller, drying was more uniform, increasing the contact area of the droplet with heat and, consequently, higher exposure of the probiotic culture to the high temperature. This caused loss of intracellular water, indispensable for the survival of the microorganism, but could also cause inactivation of proteins essential for the maintenance of cellular balance [36,39].

Liu et al. [40], when studying the influence of fermentation conditions and fatty acid composition of *Lactobacillus reuteri* I5007 membranes and their survival after the lyophilization process, observed a low survival of the microorganism, varying from 27% to 38%. The low survival rate compared to this study may have been because different species of *Lactobacillus* have different levels of tolerance to stress. Additionally, the freezing temperatures and water crystal formation within the cell can cause disruption and loss of cell concentration. When compared to our result, it suggests that even though high temperatures were used, the cell loss was lower than the lyophilization process.

#### 3.4.2. Process Yield

In addition to the viability of probiotic culture, the yield is an important parameter when analyzing the drying process and microencapsulated product quality. According to [41], the yield is related to the beneficial cost of the process because the higher the yield, the greater the economic return. The main factors for the low yield are the deposition on the walls of the dryer and the low efficiency of the cyclone to collect fine particles. Once the product is not obtained in the cyclone, it remains inside the dryer for longer periods, resulting in changes in product quality [42].

Process yields varied between 17.21% and 49.67%. A first-order polynomial model was fitted to the experimental data (Equation (9)) to better understand the impact of process variables on the process yield.The highest yield was verified when the atomization air flow rate of 4 kg/h with 0% of gelatin was used. Similar values were found by Guergoletto, Busanello, and Garcia [43], which observed a lower yield (24.04%) when using gelatin as an encapsulating agent and a yield of 57.07% when using maltodextrin in the drying process of the probiotic fermented pulp from juçara. The inclusion of gelatin increased the suspension viscosity, which resulted in larger droplets when the fluid was atomized. Dehydration from droplet to the particle was slower when the droplet was bigger, and therefore, the newly formed particles hit the drying chamber wall still wet, causing more product accumulation. Additionally, the atomization air flow rate presented a significant effect (*p* < 0.05) on process yield. Therefore, these results suggest that smaller droplets provide better yields.
(9)$$\mathrm{Yield}\left(\%\right)=0.53+{13.06\times F}_{\mathrm{atom}}-2.47\times G-{0.36\times F}_{\mathrm{atom}}\times G$$

#### 3.4.3. Phenolic Content

The content of phenolic compounds (PC) of the reconstituted passion fruit probiotic pulp samples ranged from 12.83 to 22.22 mg EGA/100 mL. The final phenolic content after fermentation was 24.13 mg EGA/mL, meaning the phenolic content retention ranged between 53.1% to 92.1%. Analyzing the effects of the independent variables, it was observed that both variables and the interaction significantly influenced the phenolic contents. A polynomial model was fitted to the data (*R*² = 82.85%) and the estimated regression coefficient (Equation (10)).
(10)$$\mathrm{PC}=15.825-1{.009\times F}_{\mathrm{atom}}-2.061\times G+{1.689\times F}_{\mathrm{atom}}\times G,\;R^{2}=82.85\%$$

Gelatin concentration at 4%, along with the atomization flow rate of 4 kg/h, provided better protection for phenolic contents because the maximum value was found for these conditions. However, at the same concentration of gelatin, when the atomization flow rate was at the lower level, the phenolic content retention dropped by 20%. This result revealed that crust formation is critical for phenolic retention. This assumption was based on the fact that drying kinetics was much faster for smaller droplets, and thus, water was withdrawn from the droplet surface at a higher rate, forming a gelatin crust at a higher rate as well. Moreover, when the gelatin concentration was lowered to 0%, the retention dropped drastically to 53%, revealing that maltodextrin provided less protection for phenolic compounds. The conservation of phenolic compounds, as well as other nutritional components of the food after processing, is of paramount importance because consumers are increasingly seeking products that maintain the maximum of the characteristics of in natura food.

#### 3.4.4. Moisture Content

Moisture content revealed the process inefficiency for complete moisture removal. This is an important parameter because it influences water activity and, thus product stabilization [44]. Moisture content values ranged from 7.6% to 6.4%, which is a typical range for spray-drying processes. Similar values were found by Dantas et al. [39], finding values ranging from 3.91% to 7.9% for spray-dried avocado pulp. Guergoletto, Busanello, and Garcia [43] observed lower values (2.7–4.0%) when evaluating the effects of encapsulating agents on the viability of *Lactobacillus reuteri* LR92 on juçara pulp. The difference could be attributed to the inlet temperature because our experiments were conducted at a low inlet temperature (112 °C). A first-order polynomial model was fitted to the experimental data (Equation (11)) to better understand the impact of process variables on the moisture content.
(11)$$\mathrm{MC}=7.081+{0.077\times F}_{\mathrm{atom}}-0.104\times G-{0.118\times F}_{\mathrm{atom}}\times G,\;R^{2}=78.13\%$$
where WC is the moisture content (%). 

Gelatin concentration presented a significant effect on moisture content. The increase in gelatin concentration led to a decrease in the water content of the samples. This increase could simply be attributed to the higher solids concentration on the feed solution. Atomization flow rate did not have an impact on the moisture content, revealing droplet size did not rule the final moisture content. This result suggest that the inlet temperature rules moisture content. According to Heidebach, Först, and Kulozik [45], the maximum recommended value to ensure prolonged stability during storage of the powder, as well as the viability of the encapsulated probiotic microorganism, is 4%. Moreover, when considering fruit juices, the diffusion rate of water is higher due to the presence of low molecular weight sugars, producing less water-binding energy [39]. For a better assessment of stability, tests such as water activity, moisture sorption isotherms, and glass transition temperature during a shelf life study are recommended. Therefore, our result suggests that higher inlet temperature should be used to reduce the final moisture content to stabilize the powder product further.

## 4. Conclusions

Probiotic passion fruit pulp from *Passiflora edulis* Sims f. *flavicarpa* Deg. fruit was successfully produced using *Lactobacillus reuteri*, qualifying it as a good medium for probiotic culture when the temperature of fermentation of 30 °C was used. The presence of phenolic compounds, along with other acidic molecules, could have been used as a second metabolic pathway and as parameters to evaluate the quality of the probiotic produced. A combination of the probiotic properties with bioactive compounds could bring health benefits for its consumers. Product stabilization was also sucessfully studied. Dehydration of probiotic passion fruit pulp and conversion into powder was achieved using spray drying. The use of gelatin in combination with maltodextrin was proven to yield better cellular viability and phenolic compound retention than the maltodextrin alone, possibly because of the higher molecular weight and lower gelification concentration of gelatin. According to the results, encapsulation materials play a more important role than drying kinetics.

## Figures and Tables

**Figure 1 foods-09-00335-f001:**
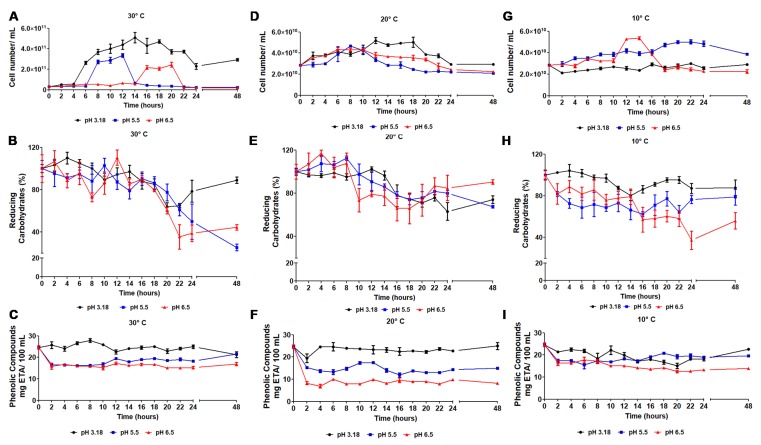
Time course of *Lactobacillus reuteri* fermentation: cell concentration at (**A**) 30 °C, (**D**) 20 °C and (**G**) 10 °C; reducing carbohydrates at (**B**) 30 °C, (**E**) 20 °C, and (**H**) 10 °C; phenolic compounds at (**C**) 30 °C, (**F**) 20 °C and (**I**) 10 °C.

**Figure 2 foods-09-00335-f002:**
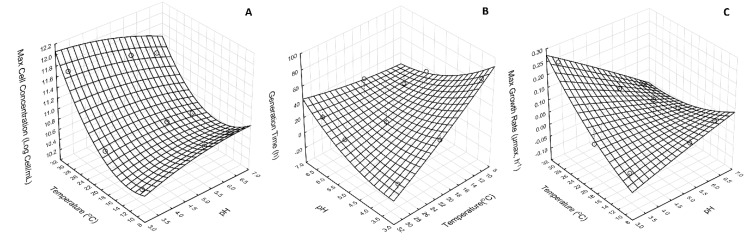
Effect of temperature and pH on (**A**) maximum cell concentration (log cell/mL), (**B**) generation time (h), and (**C**) max cell growth (h^−1^).

**Figure 3 foods-09-00335-f003:**
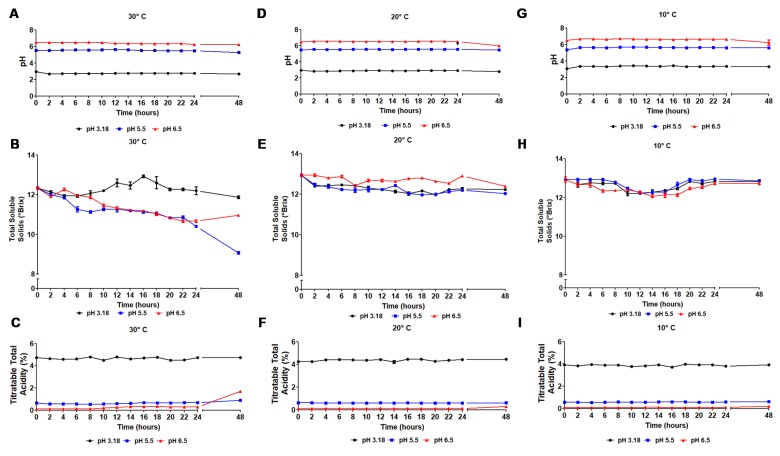
Time course of *Lactobacillus reuteri* fermentation: pH at (**A**) 30 °C, (**D**) 20 °C and (**G**) 10 °C; total soluble solids (°Brix) at (**B**) 30 °C, (**E**) 20 °C and (**H**) 10 °C; titratable total acidity(%) (**C**) 30 °C, (**F**) 20 °C and (**I**) 10 °C.

**Figure 4 foods-09-00335-f004:**
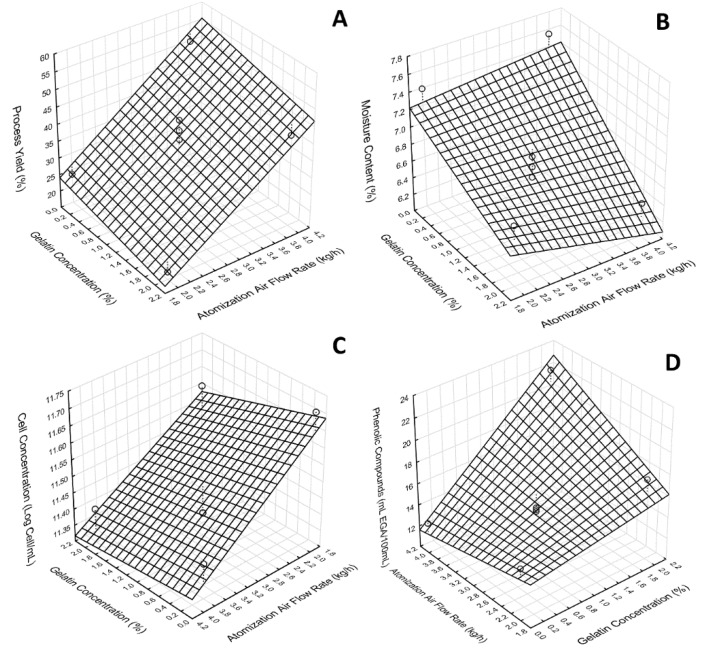
Effect of atomization flow rate and gelatin concentration on (**A**) process yield (%), (**B**) moisture content (%), (**C**) cell concentration, (**D**) phenolic content (mg EGA/100 mL).

**Table 1 foods-09-00335-t001:** Results from the fermentaion of yellow passion fruit pulp (*Passiflora edulis* Sims f. *flavicarpa* Deg.) using *Lactobacillus reuteri* under different conditions.

Parameter	In Natura Pulp	10 °C			20 °C			30 °C		
		3.18	5.5	6.5	3.18	5.5	6.5	3.18	5.5	6.5
pH	3.18 ± 0.01	3.36 ± 0.01	5.61 ± 0.01	6.64 ± 0.01	2.92 ± 0.01	5.56 ± 0.01	6.22 ± 0.01	2.74 ± 0.01	5.46 ± 0.01	6.23 ± 0.01
Total titratable acidity (%)	4.55 ± 0.06	4.55 ± 0.06	4.55 ± 0.06	4.55 ± 0.06	4.44 ± 0.01	0.60 ± 0.01	0.12 ± 0.01	4.73 ± 0.05	0.69 ± 0.01	0.30 ± 0.01
Total soluble solids (°Brix)	12.33 ± 0.12	12.03 ± 0.03	12.07 ± 0.03	12.63 ± 0.17	12.27 ± 0.07	12.20 ± 0.12	12.09 ± 0.06	12.20 ± 0.20	10.40 ± 0.05	10.367 ± 0.07
Reducing sugars (%)	4.86 ± 0.59	87.11 ± 4.93	76.44 ± 3.97	37.31 ± 8.84	63.07 ± 8.13	80.03 ± 2.75	84.87 ± 11.78	78.61 ± 9.89	49.89 ± 9.52	38.91 ± 7.69
Phenolic compounds (mg EGA.100 mL^−1^)	36.56 ± 7.89	18.28 ± 1.72	19.06 ± 0.45	13.23 ± 0.21	22.69 ± 0.74	14.98 ± 0.64	9.91 ± 0.45	24.98 ± 1.45	18.28 ± 0.56	15.30 ± 1.06
Proteins(mg BSA.mL^−1^)	0.62 ± 0.15	0.66 ± 0.14	0.55 ± 0.16	0.78 ± 0.09	0.52 ± 0.09	0.54 ± 0.08	0.66 ± 0.05	1.42 ± 0.09	1.52 ± 0.17	1.43 ± 0.08
Generation time (h)	NA	76.74	45.61	12.59	27.99	7.16	45.69	2.68	13.40	23.93
Max cell concentration (cells/mL)	NA	3.01 × 10^10^	5.30 × 10^10^	5.51 × 10^10^	5.10 × 10^10^	5.29 × 10^10^	4.67 × 10^10^	5.59 × 10^11^	3.68 × 10^11^	2.55 × 10^11^
Max growth rate (μ_max_ h^−1^)	NA	0.009	0.015	0.055	0.025	0.097	0.015	0.026	0.052	0.029

NA: not available.

**Table 2 foods-09-00335-t002:** Spray-drying process conditions and experimental results of the probiotic passion fruit pulp powder.

	Units	1	2	3	4	5	6	7
Atomization Flow tate	(kg/h)	2.0	2.0	4.0	4.0	3.0	3.0	3.0
Gelatin Concentration	(%)	0.0	2.0	0.0	2.0	1.0	1.0	1.0
Process yield	(%)	23.6	17.22	49.67	41.85	37.32	42.99	39.94
Moisture Content	(%)	7.39 ± 0.34a	6.71 ± 0.20ab	7.54 ± 0.26a	6.39 ± 0.34c	6.53 ± 0.34c	6.77 ± 0.32ac	6.65 ± 0.25ac
Cell viability	Log cells mL^−1^	11.72 ± 0.007a	11.62 ± 0.039b	11.45 ± 0.034c	11.4 ± 0.041c	11.41 ± 0.038c	11.41 ± 0.029c	11.41 ± 0.007c
Phenolic content	mg EGA/100 mL	14 ± 1.5a	17 ± 2.8ab	12.83 ± 0.15a	22.0 ± 2.8c	14.59 ± 0.61a	14.24 ± 0.52a	14.41 ± 0.54a

Values at each row with the same combination of letters do not have significant difference according to Tukey’s test for *p* < 0.05.

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
