# Peer review of "Production of Probiotic Passion Fruit (Passiflora edulis Sims f. flavicarpa Deg.) Drink Using Lactobacillus reuteri and Microencapsulation via Spray Drying"

_foods, 2020, doi:10.3390/foods9030335_

Round 1

Reviewer 1 Report

Ref: Foods-723794: Production of probiotic passion fruit (Passiflora edulis Sims f.flavicarpa Deg.) drink using Lactobacillus reuteri and microencapsulation via Spray drying

General comments:

The aim of this study was, according to the title, the production of probiotic Passion Fruit using Lactobacillus reuteri and microencapsulation by spray drying. First, the fermentaion of yellow passion fruit pulp was performed using a commercial strain of Lactobacillus reuteri under different conditions.

Major concerns:

Lines 63-64: the combination of high cell viability with high bioactive compounds retention can also be challenging for a spray drying process. Comments: The authors rightly stress the importance of maintaining cellular viability during the production of a probiotic powder via spray-drying. The cell survival was calculated on the number of viable cells (lines 116-117). In Table 2, results on cell viability were expressed as Log CFU per mL (CFU= Colony forming units) However, the cell concentration was determined by direct counting in Neubauer's chamber (line 87). How was the viable count determined? Is it only by direct count or by plating on a culture medium? Please justify or modify?

Lines 70-71: The culture of Lactobacillus reuteri was purchased from the company Official Pharmacy. Comments: To consider a strain as probiotic or potentially probiotic, it is essential to provide accurate identification of the strain. Usually, a probiotic strain has a code and has been deposited in a culture collection. Without this information, the strain cannot be recognized as a probiotic. In this study, the strain is unknown and it is not possible to identify in the literature, studies supporting these health benefits. In addition, the source of the supplier is not described(city, country). Please provide more information.

Specific comments:

Line 160: Please define EGA : equivalent of galleic acid.

Table 1, Table 2, Figure 4 : please replace ETA by EGA

Minor comments:

Line 12: Please replace microflora by microbiota

Line 63: Please replace viabiilty by viability

Author Response

February  15, 2020

Dr. Christopher J. Smith

Editor in Chief

Foods

Dear Dr. Christopher J. Smith

Thank you for the opportunity to revise the manuscript. We appreciate the detailed comments that were received to help us improve the manuscript. We believe we have now addressed all of the reviewers’ concerns and that the manuscript is ready for resubmission. All changes are highlighted in yellow in the text.

Below is a point-by-point response:

Reviewers' comments:
Reviewer #1:

Point 1 – Line 63-64: the co: the combination of high cell viability with high bioactive compounds retention can also be challenging for a spray drying process. Comments: The authors rightly stress the importance of maintaining cellular viability during the production of a probiotic powder via spray-drying. The cell survival was calculated on the number of viable cells (lines 116-117). In Table 2, results on cell viability were expressed as Log CFU per mL (CFU= Colony forming units) However, the cell concentration was determined by direct counting in Neubauer's chamber (line 87). How was the viable count determined? Is it only by direct count or by plating on a culture medium? Please justify or modify?

Response 1:  Thank you for your suggestion. We have now addressed this question through the corrections made in the manuscript. We corrected to cell number.

Point 2 - Lines 70-71: The culture of Lactobacillus reuteri was purchased from the company Official Pharmacy. Comments: To consider a strain as probiotic or potentially probiotic, it is essential to provide accurate identification of the strain. Usually, a probiotic strain has a code and has been deposited in a culture collection. Without this information, the strain cannot be recognized as a probiotic. In this study, the strain is unknown and it is not possible to identify in the literature, studies supporting these health benefits. In addition, the source of the supplier is not described(city, country). Please provide more information.

Response 2: We have now addressed this question through the changes made in the manuscript. In the methods section was added the strain of the Lactobacillus reuteri

Point 3 – Line 160: Please define EGA : equivalent of galleic acid.

Response 3: We apologize for the error, and this issue has now been rectified.

Point 4 – Table 1, Table 2, Figure 4 : please replace ETA by EGA

Response 4: We apologize for the error, and this issue has now been rectified.

Point 5 – Line 12: Please replace microflora by microbiota

Response 5: We apologize for the error, and this issue has now been rectified.

Point 6 –  Line 63: Please replace viabiilty by viability

Response 6: We apologize for the error, and this issue has now been rectified.

Yours truly,

Matheus A. de B Pasquali & Co-Authors

Matheus Augusto de Bittencourt Pasquali M.Sc., Ph.D.

Adjunct Professor, Department of Food Engineering

Federal University of Campina Grande

Street Aprigio Veloso, 882 – Phone + 55 83 2101 1986

Campina Grande – PB-Brazil,  Zip Code 58429900  

Reviewer 2 Report

The originallity of the article is good and the analytical part very good. However the outcome is not suppoorted very well. The language and expressions need to be improved. I believe that the authors should revise specific points in the text where the each conclusion is discussed in order to be more understood from teh readers. For instance they should clarify better the applications of the product. The form of the product (wet form, dry form).

Specific comments

Lines 64-65: The sentence is confusing. Please rephrase

Line 138: Delete “the”

Lines 142-144: The sentence is confusing. Please rephrase

Line 146: Please rephrase to: mostly because it is considered an innovative product.

Line 149: Change uses to utilizes

Lines 152-154: The sentence is confusing. Please rephrase

Lines 154-157: The sentence is confusing. Please rephrase

Lines 163-164: The sentence is confusing. Please rephrase

Line 172: Add a reference

Line 184, 187: Change pHs to pH values

Lines 188-189: The sentence is confusing. Please rephrase

Lines 192-195: The sentence is confusing. Please rephrase

Lines 217-219: The sentence is confusing. Please rephrase

Line 267: Add a reference

Author Response

February  15, 2020

Dr. Christopher J. Smith

Editor in Chief

Foods

Dear Dr. Christopher J. Smith

Thank you for the opportunity to revise the manuscript. We appreciate the detailed comments that were received to help us improve the manuscript. We believe we have now addressed all of the reviewers’ concerns and that the manuscript is ready for resubmission. All changes are highlighted in yellow in the text.

Below is a point-by-point response:

Reviewers' comments:

Reviewer #2

Point 1 - “The originallity of the article is good and the analytical part very good. However the outcome is not suppoorted very well. The language and expressions need to be improved. I believe that the authors should revise specific points in the text where the each conclusion is discussed in order to be more understood from teh readers. For instance they should clarify better the applications of the product. The form of the product (wet form, dry form).”

Response 1: We agree these suggestions would be informative.  Thank you for your suggestion. We have now addressed this question through the corrections made in the manuscript.

Point 2 Lines 64-65: The sentence is confusing. Please rephrase

Response 2: We have now addressed this question through the changes made in the manuscript.

Point 3 Line 138: Delete “the”

Response 3: We apologize for the error, and this issue has now been rectified.

Point 4 Lines 142-144: The sentence is confusing. Please rephrase

Response 4: We have now addressed this question through the changes made in the manuscript.

Point 5 Line 146: Please rephrase to: mostly because it is considered an innovative product.

Response 5: We apologize for the error, and this issue has now been rectified.

Point 6 Line 149: Change uses to utilizes

Response 6: We apologize for the error, and this issue has now been rectified.

Point 7 Lines 152-154: The sentence is confusing. Please rephrase

Response 7: We have now addressed this question through the changes made in the manuscript.

Point 8 Lines 154-157: The sentence is confusing. Please rephrase

Response 8: We have now addressed this question through the changes made in the manuscript.

Point 9 Lines 163-164: The sentence is confusing. Please rephrase

Response 9: We have now addressed this question through the changes made in the manuscript.

Point 10 Line 172: Add a reference

Response 10: We have now addressed this question through the changes made in the manuscript, reference was added.

Point 11 Line 184, 187: Change pHs to pH values

Response 11: We apologize for the error, and this issue has now been rectified.

Point 12 Lines 188-189: The sentence is confusing. Please rephrase

Response 12: We have now addressed this question through the changes made in the manuscript.

Point 13 Lines 192-195: The sentence is confusing. Please rephrase

Response 13: We have now addressed this question through the changes made in the manuscript.

Point 14 Lines 217-219: The sentence is confusing. Please rephrase

Response 14: We have now addressed this question through the changes made in the manuscript. The sentence was withdrawn. 

Point 15 Line 267: Add a reference

Response 15 - We have now addressed this question through the changes made in the manuscript, reference was added.

Yours truly,

Matheus A. de B Pasquali & Co-Authors

Matheus Augusto de Bittencourt Pasquali M.Sc., Ph.D.

Adjunct Professor, Department of Food Engineering

Federal University of Campina Grande

Street Aprigio Veloso, 882 – Phone + 55 83 2101 1986

Campina Grande – PB-Brazil,  Zip Code 58429900  

Round 2

Reviewer 1 Report

No additional comments